# Disentangling the Effects of Data Augmentation and Format Transform in Self-Supervised Learning of Image Representations

**Neha Kalibhat**
University of Maryland, College Park
nehamk@umd.edu

**Warren Morningstar**
Google Research

**Alex Bijamov**
Google Research

**Luyang Liu**
Google Research

**Karan Singhal**
Google Research

**Philip Mansfield**
Google Research

**Editors:** Marco Fumero, Clementine Domine, Zorah Lähner, Donato Crisostomi, Luca Moschella, Kimberly Stachenfeld

## Abstract

Self-Supervised Learning (SSL) enables training performant models using limited labeled data. One of the pillars underlying vision SSL is the use of data augmentations—perturbations of the input which do not significantly alter its semantic content. For audio and other temporal signals, augmentations are commonly used alongside format transforms such as Fourier transforms or wavelet transforms. Unlike augmentations, format transforms do not change the information contained in the data; rather, they express the same information in different coordinates. In this paper, we study the effects of format transforms and augmentations both separately and together on vision SSL. We define augmentations in frequency space called Fourier Domain Augmentations (FDA) and show that training SSL models on a combination of these and image augmentations can improve the downstream classification accuracy by up to $1.3\%$ on ImageNet-1K. We also show improvements against SSL baselines in few-shot and transfer learning setups using FDA. Surprisingly, we also observe that format transforms can improve the quality of learned representations even without augmentations; however, the combination of the two techniques yields better quality.

## 1 Introduction

In the fast-evolving landscape of deep learning and computer vision, self-supervised learning has emerged as a powerful paradigm for foundation models [1, 2]. Its success is rooted in its ability to learn robust and generalizable representations from unlabelled data with no supervision. Existing SSL approaches have been categorized into two main types: generative [3] and invariance-based [4, 5, 6, 2, 7, 8]. The latter involves joint-embedding pre-training with two or more *views* of the same input data sample. To prevent joint-embedding representations from collapsing (converge to identical representations) during pre-training, it is crucial to employ stochastic augmentations like random crop, color jitter, Gaussian blur, solarization etc. These augmentations are often hand-crafted for specific downstream tasks and may not transfer well to other tasks [9, 10]. We show evidence (Figure 2) that progressively adding more hand-crafted augmentations improves downstream linear probing performance and conversely, removing any given augmentation always hurts performance among 3

Proceedings of the I edition of the Workshop on Unifying Representations in Neural Models (UniReps 2023).

self-supervised baselines. We therefore hypothesize that increasing augmentation diversity during pre-training allows representations to become invariant to more nuisance concepts and could improve downstream linear probing performance.

Meanwhile, in the audio and speech domain, recent works [11, 12, 13] have successfully performed self-supervised learning by maximizing the mutual information between time and frequency formats in the latent space using the Fourier Transform and a small number of format-specific augmentations. This mode of transformation allows us to represent the same data under different coordinates. This is unlike hand-crafted augmentations, since the data remains unperturbed. Prior works [14, 15, 16, 17] have utilized the Fourier space to unify multi-domain latent spaces to benefit tasks like domain generalization and image-to-image translation. We use the term *format transform* and *Fourier transform* interchangeably in the context of images.

In this paper, we integrate both notions presented above. We study the effect of incorporating augmentations in the Fourier domain of images with the goal of increasing overall augmentation diversity. To this end, we propose a pipeline of augmentations called **Fourier Domain Augmentations (FDA)** that can be applied in the complex Fourier domain. When data after these FDAs are inverted back to the image space, we observe that they produce unique textures and patterns, which cannot be easily reproduced by directly perturbing the image space.

We study the combined effect of applying FDA along with standard image augmentations on pre-training state-of-the-art self-supervised baselines including SimCLR [4], BYOL [7], MoCov2 [18] and SimSiam [5] on ImageNet-1K [19]. We show an average improvement of $1\%$ in the top-1 accuracy during downstream linear probing. We also evaluate other downstream tasks including few-shot learning and transfer learning and show qualitative improvements on image retrieval with the use of FDAs.

Our results confirm our initial hypothesis of the need for augmentation diversity. We perform ablations where we study the independent effects of augmentations in the image space and the frequency space in a single-encoder contrastive learning setup (SimCLR). We explore the results of maximizing agreement between two augmented views where the augmentation can be any one of (i) standard image augmentations (ii) Fourier-mode augmentations and (iii) the combination of both.

Finally, we examine the individual effect of using the format transform itself disentangled from any augmentations. This experiment is to understand if self-supervised learning can benefit from encoding images presented in multiple formats without any augmentations i.e., the raw image and Fourier transform. To achieve this we design a dual-encoder setup with contrastive learning where each encoder is exposed to one modality, either raw image or Fourier image. We observe that providing the Fourier transform as one of the views during pre-training improves linear probing performance by $16\%$ compared to raw image pre-training in lieu of any augmentations. We further explore the benefit of augmentations (both image and frequency) in this dual-encoder setup. Across all ablations, we observe that combining image and FDA while pre-training in the image domain results in the best downstream performance.

## 2   Background

**Self-Supervised Learning** is a powerful approach of learning representations from large amounts of data without the use of labels. Learned representations can later be used for downstream tasks [20] directly or with inexpensive fine-tuning. Representations are learned by solving *pretext tasks* which can involve predicting simple transformations on a given image like rotations [21], jigsaw [22] or color [23]. However, more successful self-supervised approaches involve joint-embedding methods which force latent space similarity between multiple augmented views of the same image sample. This can be achieved via contrastive or InfoNCE loss [24, 4, 18, 5], self-distillation [2, 7] or by redundancy reduction in the latent space [8, 6]. Regardless of the training paradigm, all joint-embedding methods rely on powerful data augmentations to control the degree of invariance beneficial for downstream tasks.

**Augmentations in Self-Supervised Learning** engender invariances which in turn introduce good inductive biases for downstream tasks [10, 25]. However, for any given downstream task, specific augmentations may be better suited over others [10, 9]. This property tends to restrict the generalization capability of many self-supervised models as using an inappropriate set of augmentations

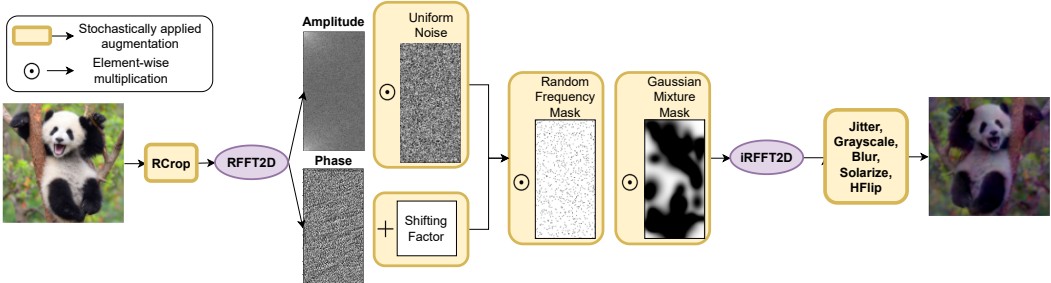

Figure 1: **Diversifying image augmentations with Fourier Domain Augmentations (FDA):** We show the pipeline of applying Fourier Mode Augmentations integrated with standard image augmentations like random cropping, color jitter, grayscale etc. We use RFFT2D (available in PyTorch and TensorFlow) to transform a random resized crop image into the Fourier space. Here, we stochastically apply *amplitude rescale*, *phase-shift*, *random frequency mask* and *Gaussian mixture mask* which together constitute Fourier Domain Augmentations (FDA). The remaining image augmentations are applied after inverting the augmented Fourier spectrum back to the image space using iRFFT2D.

can significantly hurt downstream performance. Therefore, a standard protocol followed by most self-supervised approaches is to identify optimal augmentations for best downstream linear probing performance on ImageNet-1K.

**Fourier-based Methods in Audio:** Self-Supervised learning has shown success in the audio/speech domain [26, 27] in predicting embeddings of future audio samples from a sequence of prior embeddings, by comparing with a context embedding derived from the sequence. Recently, Wang et. al [11] have extended these results by directly comparing two augmented versions of a given audio sample rather than utilizing a context embedding. In their work one version of the audio sample is in the time-domain format, with augmentations directly applied to the waveform, while the other has been Fourier-transformed into the frequency-domain, with augmentations applied to the spectrogram. Encoders for the two formats are simultaneously trained so that their output embedding vectors align when they arise from the same data source. Specifically, time-domain augmentations involve masking (removing) some time intervals and adding noise. Frequency-domain augmentations involve masking (removing) some frequency intervals and shifting all frequencies by an integer constant. [12] also did contrastive learning on representations of two different signal formats; namely a waveform (not necessarily audio), and a scaleogram arising from a wavelet transform. However, no data augmentations were explored in that work. [13] train a joint time-frequency representation, where self-supervision is implemented by penalizing the distance between a signal's time and frequency representations, each pretrained contrastively.

The contrasting of multiple formats (raw and frequency) of the same input is especially interesting even in the image space, as it potentially allows generating rich embeddings that encode both modalities. To the best of our knowledge, no analysis has been done of the separate and combined effects of Fourier space augmentations and image augmentations. Moreover, neither augmentations in the Fourier space nor the direct use of Fourier space in self-supervision have been properly explored on image data for vision models.

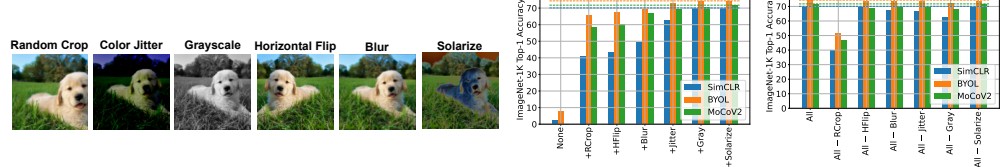

Figure 2: **Augmentation Diversity:** We display commonly used hand-crafted augmentations for self-supervised learning on the left. We demonstrate the effect of increasing diversity in pre-training augmentations (first plot) and removing individual augmentations (second plot). The best performance is retained when all given augmentations are used in 3 baselines.

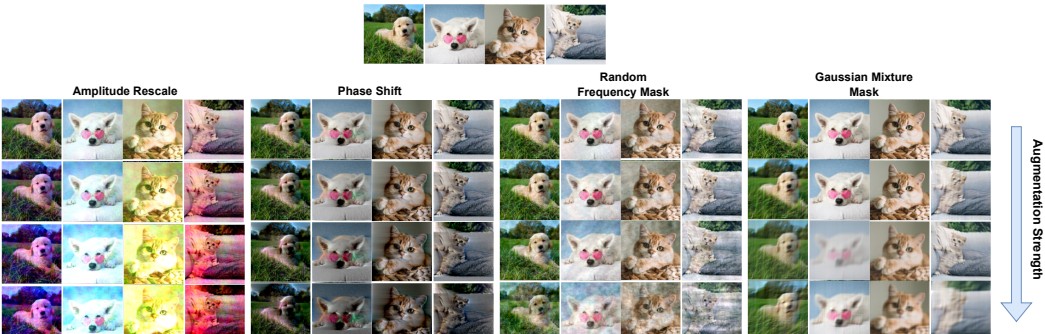

Figure 3: **Fourier Domain Augmentations (FDA):** We illustrate the result of applying each augmentation in FDA when inverted back into the image domain - amplitude rescale, phase shift, random frequency mask, Gaussian mixture mask. We vary the strength using augmentation-specific hyper-parameters $m, n, p, q, k, o$ (see Section 4). We tune these hyper-parameters (no training required) such that images are perturbed sufficiently without hiding the core ground-truth attributes.

# 3 Importance of Diversity in Pre-Training Augmentations

In this section, we illustrate the strong dependence that joint-embedding self-supervised models have on pre-training augmentations. We hypothesize that each augmentation tackles a specific type of *invariance*. Depending on the downstream task, a model's generalization power can be improved by enforcing invariance to physical properties irrelevant to the ground truth [10, 25, 9]. The standard set of augmentations used by self-supervised models are - random cropping and resizing, horizontal flip, color jittering, grayscale, Gaussian blurring and solarization. We display an example of these augmentations in Figure 2 (top panel). These augmentations have been hand-crafted to show competitive performance in downstream classification, particularly on ImageNet-1K.

In Figure 2 (bottom left plot), we show the effect of progressively adding individual augmentations while pre-training SimCLR, BYOL and MoCov2 on ImageNet-1K and measuring the linear probing accuracy. Each baseline demonstrates the best performance when all of the above augmentations are used. This result supports our claim of diversity playing an important role in producing easily classifiable representations.

While the diversity of augmentations is necessary, it is also important that each augmentation attacks specific invariances. In Figure 2 (bottom, right plot), we show the impact of removing individual augmentations while maintaining the rest. Each model shows a drop in performance when any of the augmentations are removed. Among these, removing random cropping shows the strongest reduction in performance (followed by grayscale) compared to the baselines which retain all augmentations.

It is important to note that regardless of the pre-training paradigm, self-supervised models only demonstrate state-of-the-art performance when all of the above augmentations are used. As more augmentations are incorporated during pre-training, the downstream performance steadily improves. This begs the question - can we additionally incorporate new augmentations to further improve linear classification performance? While most of the proposed augmentation strategies [6, 22, 28, 29] perturb the image directly, we shift the focus to leverage the format transform of images to incorporate new information and invariances. We first explore these benefits by augmenting the Fourier spectrum and returning to the image space via an inverse transform. We then explore utilizing the Fourier spectrum directly in joint-embedding pre-training to study its independent effect.

# 4 Fourier Domain Augmentations (FDA)

The Discrete Fourier Transform of a single-channel 2-dimensional image $\mathbf{x} \in \mathbb{R}^{H \times W}$ is given by,

$$\mathcal{F}(\mathbf{x})_{u,v} = \sum_{h=0}^{H-1} \sum_{w=0}^{W-1} e^{-2\pi i \left( \frac{h}{H} u + \frac{w}{W} v \right)} x_{h,w} \tag{1}$$

where, $u = \{0...H - 1\}$ and $v = \{0...W - 1\}$. The Fourier transform can be applied over every image channel (RGB). Both $\mathcal{F}$ and $\mathcal{F}^{-1}$ can be computed efficiently using the Fast Fourier Transform algorithm [30]. Since the FFT of a real signal is Hermitian-symmetric, we use the RFFT2D operation (provided by PyTorch and TensorFlow), which provides only the positive frequency terms to avoid redundancy.

Let $\mathbf{f}$ denote the complex-valued Fourier spectrum of the image $\mathbf{x}$ (Equation 1). The real and imaginary components of $\mathbf{f}$ are denoted by $\mathcal{R}(\mathbf{f}) = \mathcal{A}(\mathbf{f})\cos\mathcal{P}(\mathbf{f})$ and $\mathcal{I}(\mathbf{f}) = \mathcal{A}(\mathbf{f})\sin\mathcal{P}(\mathbf{f})$ respectively where, $\mathcal{A}(\mathbf{f})$ is the amplitude and $\mathcal{P}(\mathbf{f})$ is the phase of the spectrum. Conversely, $\mathcal{A}(\mathbf{f}) = \sqrt{\mathcal{R}^2(\mathbf{f}) + \mathcal{I}^2(\mathbf{f})}$, and $\mathcal{P}(\mathbf{f}) = atan2\,(\mathcal{I}(\mathbf{f}), \mathcal{R}(\mathbf{f}))$.

The Fourier spectrum provides a number of unique insights into the image signal. A well-known and often exploited property [31, 32, 33, 34] is that the amplitude represents low-level statistics and superficial patterns in the image while the phase preserves structural and semantic information. Traditional image processing techniques [35] involved using a circular kernel mask on the Fourier spectrum to turn off high-frequency modes (low-pass filter) to create a *blurring* effect, after inverting back to the image space ($\mathcal{F}^{-1}$). On the other hand, turning off low-frequency modes (high-pass filter) creates a *sharpening* effect. Inverting the Fourier spectrum back to the image space lets us apply our method as new augmentations in addition to standard image augmentations and does not require us to re-define the self-supervised training pipeline. In Section 6, we study disentangle the effect of format transform and augmentations with the use of a designated image encoder and frequency encoder where we directly encode Fourier input ($\mathbf{f}$) into representations.

We propose the following general-purpose format transformations that perturb different properties in the Fourier spectrum, producing unique augmentations when inverted back to the image space.

- **Amplitude Re-scale:** We prepare a uniform noise vector $\mathbf{p} \in \mathbb{R}^{H \times W}$ within a range $[m, n]$ where, $m, n > 0$ (selected empirically). We multiply this noise with the amplitude of the spectrum,

$$\mathcal{A}(\mathbf{f}) = \mathcal{A}(\mathbf{f}) \odot \mathbf{p}$$

  A randomly sampled noise is applied to each channel of the FFT of the 3-channel image. When this augmentation is inverted to the image domain ($\mathcal{F}^{-1}$), it results in non-uniform perturbations to the image color scope.

- **Phase Shift:** We randomly sample a constant *shifting factor* $\theta \in \mathbb{R}$ within the range $[p, q]$ where, $p, q > 0$ (selected empirically). The phase is shifted as follows,

$$\mathcal{P}(\mathbf{f}) = \mathcal{P}(\mathbf{f}) \pm \theta$$

  This transform brings about a *movement effect* in the image wherein certain high-frequency attributes are brightened.

- **Random Frequency Mask:** We define a binary mask $\mathbf{h}$, commonly across all channels where $k\%$ of frequencies are set to 0. We also ensure that the zero frequency mode ($h_{0,0}$) is always enabled so that semantic information is largely retained.

$$\mathbf{f} = \mathbf{f} \odot \mathbf{h}$$

  This transform randomly turns off both high and low frequency modes across all channels. This preserves the color scope but results in a unique *cloudy* texture non-uniformly applied across the image.

- **Gaussian Mixture Mask:** Unlike, low-pass and high-pass filters which apply a single circular kernel at the center of the spectrum, we propose a more general form of frequency-band masking. We prepare a Gaussian Mixture Mask with a randomly sampled set of origins, $\mathbf{c} \in \mathbb{R}^{o \times 2}$ and standard deviations, $\sigma \in \mathbb{R}^{o \times 2}$. We draw a 2D Gaussian kernel around each origin given by,

$$\mathcal{G}(u, v, \mathbf{c}, \sigma) = \exp -\left(\frac{(u - o_0)^2}{2\sigma_0^2} + \frac{(v - o_1)^2}{2\sigma_1^2}\right)$$

  An illustration of the resulting mask is shown in Figure 1. This method flexibly masks low and high frequencies and the resulting images show unique textures containing both blurred and sharpened artifacts.

Figure 3 illustrates each proposed augmentation on a common set of images. We vary the strength of each augmentation via their respective hyperparameters (including $m, n, p, q, k, o$). Each augmentation's strength can be tuned such that it introduces sufficient invariance but does not obfuscate the main content of the image relevant to the ground truth (in the downstream task). More importantly, we confirm this effect when each augmentation is used together with other FDA or image augmentations. Note that this is a subjective process involving visual examination of images. Due to resource constraints, we apply the same set of augmentation parameters for all our experiments (detailed in the Appendix) however, these can be further tuned for each specific baseline. In the next section, we perform pre-training experiments on a combination of both FDA and image augmentations following the pipeline illustrated in Figure 1.

## 5 Experimental Results

### 5.1 Experimental Setup

We examine 4 self-supervised baselines including SimCLR [4], MoCov2 [18], BYOL [7] and SimSiam [5]. Our TensorFlow [36] implementation replicates the training paradigms of each model including their encoder architecture (projector, predictor, momentum encoder etc.), loss, learning rate scheduling (cosine anneal) and optimizer (LARS [37]). More details about training detailed in the Appendix. We use the ResNet-50 [38] backbone for all our experiments. To be consistent, we apply the following image augmentations across all baselines - random resized crop, color jitter, horizontal flip, Gaussian blur, grayscale and solarize. Within this augmentation pipeline, we incorporate our

Table 1: **ImageNet-1K Pre-Training with FDA:** We report the linear probing top-1 accuracy of 4 self-supervised baselines pre-trained on ImageNet-1K. When FDA is applied in addition to standard image augmentations, we observe $\sim 1\%$ improvement in performance across all models. We report the mean and standard deviation across 3 random seeds.

| | *Top-1 Accuracy - ImageNet-1K* | | | |
| | **SimCLR** | **BYOL** | **MoCo v2** | **SimSiam** |
| --- | --- | --- | --- | --- |
| **Baseline** | 69.2 (0.3) | 74.3 (0.5) | 71.7 (0.7) | 73.7 (0.2) |
| **+ FDA (Ours)** | **70.5** (0.1) | **74.7** (0.6) | **73.0** (0.4) | **74.3** (0.5) |

Fourier Domain Augmentations (FDA) as shown in Figure 1. All other training details and hyperparameters are mentioned in the Appendix. SimCLR follows a single-encoder setup while MoCo, BYOL and SimSiam use a dual-encoder setup where one of the encoders is used for downstream tasks. We find that applying FDA to only the left view (left encoder is used for downstream tasks) in addition to existing image augmentations provides the best results as opposed to applying on both views. We perform standard linear probing for evaluation where we train a linear classifier on frozen pre-trained representations.

### 5.2 ImageNet Pre-Training

We pre-train SimCLR, BYOL, MoCov2 and SimSiam on ImageNet-1K [19] by further diversifying the left view image augmentations with FDA. In Table 1, we summarize the linear probing top-1 accuracy for each model compared to their baselines which do not use FDA. We observe that FDA shows as average improvement of $\sim 1\%$ with the highest improvement in MoCo v2 of $1.3\%$. Recall Figure 2 where we demonstrated the steady improvement in downstream performance as more augmentations are added. Our improvements with FDA solidifies our initial claims about the importance of diversity.

### 5.3 Transfer and Few-Shot Learning

We perform few-shot and transfer learning on the above frozen ImageNet pre-trained self-supervised baselines. In the few-shot setup, we apply 5-shot and 10-shot learning regimes where the training set contains 5 or 10 images per label respectively. We test for transfer learning on iNaturalist (5089 classes) [39], DomainNet Painting (345 classes) [40], Food101 (101 classes) [41] and Places365 (400 classes) [42]. We observe that pre-training with FDA largely benefits both few-shot and transfer learning tasks across all baselines. We observe the highest average improvement in MoCo.

Table 2: **Transfer and few-shot learning with FDA pre-trained encoders:** We evaluate the few-shot (5-shot, 10-shot) and transfer learning performance on the frozen encoder from Table 1. We observe that baselines pre-trained with FDA improve the top-1 accuracy over most setups.

| | ImageNet (5-shot) | ImageNet (10-shot) | iNaturalist | DomainNet (Painting) | Food101 | Places365 | Average |
|---|---|---|---|---|---|---|---|
| **SimCLR** | 38.2 | 45.6 | 45.9 | 61.8 | 73.7 | 50.4 | 52.6 |
| **+ FDA** | **39.3** | **46.6** | **47.7** | **63.6** | **74.4** | **51.0** | **53.8** |
| **BYOL** | 47.5 | **54.2** | 52.4 | 67.1 | **77.0** | 50.6 | 58.1 |
| **+ FDA** | **47.6** | 53.7 | **52.5** | **68.0** | 76.7 | **51.1** | **58.3** |
| **MoCo v2** | 43.5 | 51.4 | 46.7 | 63.9 | 75.4 | 51.1 | 55.3 |
| **+ FDA** | **43.8** | **52.8** | **48.6** | **64.0** | **76.8** | **51.9** | **56.2** |
| **SimSiam** | 46.2 | 53.3 | 51.5 | 66.7 | 76.5 | **50.3** | 57.4 |
| **+ FDA** | **46.2** | **53.5** | **52.7** | **67.3** | **76.7** | 50.1 | **57.8** |

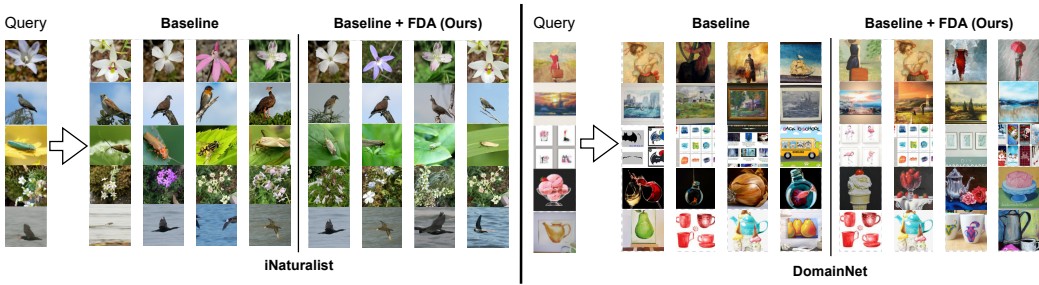

Figure 4: **Image retrieval:** We test the image retrieval quality of vanilla MoCov2 and MoCov2 pre-trained with FDA on ImageNet on transfer datasets, iNaturalist and DomainNet. We observe that the top retrieved images in MoCo FMA visibly match the semantics of the query image better.

## 5.4 Image Retrieval on Transfer Datasets

We also employ image retrieval as a qualitative evaluator of the learned representations. Given a query image, we retrieve the top-4 nearest neighbours in the representation space using cosine similarity as the distance metric. Specifically, given a sample $\mathbf{x}$ we retrieve $\arg\max 4_{\mathbf{y}} \frac{\mathbf{x}^T \mathbf{y}}{\|\mathbf{x}\|\|\mathbf{y}\|}$ from the dataset. In Figure 4, we display these results on 5 query images each from the test set of iNaturalist and DomainNet and compare the retrieved neighbours between MoCov2 baseline and MoCov2 trained with FDA on ImageNet-1K. The objective of this experiment is that the nearest neighbours should closely match the semantics of the retrieved images. This property is upheld in some FDA trained MoCo examples like the ice cream, teapot and woman with suitcase in DomainNet.

Table 3: **Disentangling the effect of FDA and image augmentations:** In a single-encoder contrastive learning setup, we ablate between the pair of augmentations used going from $\mathbf{x}$ (no augmentations) to $A_{im}(\mathcal{F}^{-1}(A_{freq}(\mathbf{f})))$ (FDA + image augmentations). Here $A_{im}(.)$ denotes the image augmentations (random crop, color jitter, blur etc.) and $A_{freq}(.)$ denotes FDA transforms we propose (amplitude rescale, phase shift etc.). We observe the best performance when using FDA + image augmentations one view and image augmentations alone in the second view. All setups are pre-trained on ImageNet-1K and we report the linear probing top-1 accuracy.

| | Augmentation | Left View | | | |
|---|---|---|---|---|---|
| | | $\mathbf{x}$ | $A_{im}(\mathbf{x})$ | $\mathcal{F}^{-1}(A_{freq}(\mathbf{f}))$ | $A_{im}(\mathcal{F}^{-1}(A_{freq}(\mathbf{f})))$ |
| **Right View** | $\mathbf{x}$ | 1.5 | 68.6 | 34.7 | 69.6 |
| | $A_{im}(\mathbf{x})$ | | 69.2 *(SimCLR baseline)* | 68.8 | **70.5** |
| | $\mathcal{F}^{-1}(A_{freq}(\mathbf{f}))$ | | | 38.9 | 67.8 |
| | $A_{im}(\mathcal{F}^{-1}(A_{freq}(\mathbf{f})))$ | | | | 70.4 |

# 6    Disentangling the Effects of Augmentation and Format Transform

We showed that pre-training state-of-the-art self-supervised baselines with FDA and standard image augmentations improves the linear classification performance of ImageNet-1K, its few-shot variants and various transfer learning datasets. This also confirms our initial hypothesis that more diverse augmentations ultimately benefit downstream tasks. However, a key aspect of our method is the utilization of the Fourier domain to introduce further diversity. Recall, our method involves multiple stages of transformations over a given image i.e., (i) The format transform (via FFT operation $\mathcal{F}$) (ii) Fourier Domain Augmentations (FDA) (iii) Inverse FFT operation to return to the image space ($\mathcal{F}^{-1}$) (iv) Standard image augmentations like color jittering, blur, grayscale etc. Therefore, it is essential to study the the effect of each operation independently to properly attribute the improvement in downstream performance.

We represent the raw input image as $\mathbf{x}$ and its Fourier transform $\mathcal{F}(\mathbf{x})$ as $\mathbf{f}$. We define the standard image augmentations, such as random crop, jitter, blur, as a function $A_{im}(.)$ and the FDAs as $A_{freq}(.)$. We train SimCLR in a single-encoder setup with a contrastive loss and various combinations of augmented views on ImageNet-1K. SimCLR uses the InfoNCE [24] objective to learn image representations. For every query sample, we maximize its similarity in the latent space with one positive view of the same sample and minimize the similarity with the remaining samples in the batch. The objective is as follows,

$$\max \log \frac{\exp\left(\mathrm{sim}(A(\mathbf{x}_i), A(\mathbf{x}_i))/\tau\right)}{\sum_{j=0}^{2N} \mathbb{1}_{i \neq j} \exp\left(\mathrm{sim}(A(\mathbf{x}_i), A(\mathbf{x}_j))/\tau\right)} \tag{2}$$

where $sim(\mathbf{a}, \mathbf{b}) = \frac{\mathbf{a}^T \mathbf{b}}{\|\mathbf{a}\|\|\mathbf{b}\|}$ and $A(.)$ is the stochastically applied set of augmentations. In Table 3, we test different pairs of augmentations between the positive views including, (i) $\mathbf{x}$: un-augmented and center-cropped image, (ii) $A_{im}(\mathbf{x})$, (iii) $\mathcal{F}^{-1}(A_{freq}(\mathbf{f}))$: FDA applied in the Fourier space and inverted back to the image space, (iv) $A_{im}(\mathcal{F}^{-1}(A_{freq}(\mathbf{f})))$: standard image augmentations applied on top of inverted FDA image. Due to the single-encoder contrastive learning setup, we present the results as an upper triangular matrix as swapping the views does not alter the overall objective.

Table 4: **Sequence of augmentations:** We follow the sequence of augmentations illustrated in Figure 1 where we apply FDA before any of the image augmentations (except random crop which is applied first). In this table, we test to see if applying FDA after image augmentations is beneficial. We observe comparable performance in both setups (on SimCLR pre-trained on ImageNet-1K).

| Augmentation | Left View | |
|---|---|---|
| | $A_{im}(\mathcal{F}^{-1}(A_{freq}(\mathbf{f})))$ | $\mathcal{F}^{-1}(A_{freq}(\mathcal{F}(A_{im}(\mathbf{x}))))$ |
| **Right View**   $A_{im}(\mathbf{x})$ | **70.5** | 70.4 |

We follow the SimCLR ImageNet-1K setup including the architecture, learning rate, scheduling, loss and optimizer. We define a naive baseline as the setup that uses a pair of raw un-augmented views ($\mathbf{x}$). The use of large batch sizes allows the model to contrast with a sufficient number of negative views, preventing collapse i.e., when all representations are identical. Nevertheless, this model achieves a low performance of $1.48\%$ as lack of augmentations inhibits the learning of informative representations. Keeping the right view un-augmented, we next experiment with different View 1 augmentations (first row in Table 3). We observe significant improvements with both FDA ($34.7\%$) and standard augmentations ($68.6\%$) applied individually, but the performance gains are highest when they are used together ($69.6\%$). Applying both augmentations to a single view also outperforms all methods which apply individual augmentations to both views. A similar trend is seen in the second row when we apply standard image augmentations to the right view. While we find that standard augmentations outperform FDA when applied individually, we attribute this mainly to the use of random cropping in standard augmentations, which significantly improves their performance (from $40\%$ to $69\%$).

As applying FDA in conjunction with image augmentations gives the best result, we next ablate how the order of FDA and image augmentations sequence affects the accuracy. In all previous experiments, we apply FDA before any other image augmentations (except random crop) following the sequence in Figure 1. In Table 4 we reverse this order and apply FDA after traditional image augmentations. Formally, this can be defined as $\mathcal{F}^{-1}(A_{freq}(\mathcal{F}(A_{im}(\mathbf{x}))))$. We observe a comparable performance in SimCLR ImageNet-1K with this image-augmentation-first strategy.

## 6.1 The Effect of Format Transform

We next disentangle the effect of using the Fourier transform ($\mathbf{f}$) directly as input to the self-supervised encoder. This experiment explores if we can produce better representations from input expressed in multiple formats (image and frequency) similar to the approach discussed in [11]. Since the Fourier spectrum of an image is complex-valued, it cannot be directly supplied to an image encoder. We therefore convert it to a real-valued 3 channel by re-scaling the spectrum to bring the values between $[0, 1]$ (same as image input). Since the RFFT2D output is of half the width ($\in \mathbb{R}^{H \times W/2 \times 3}$) as the image, we interleave the real and imaginary components such that the resulting frequency image is the same shape as that of the image ($\in \mathbb{R}^{H \times W \times 3}$).

Format transforms represent the information in frequency coordinates, which are incompatible with the image coordinate system. We therefore deploy a two-encoder setup where the first encoder $g_{im}(.)$ (left) only takes image input and the second $g_{freq}(.)$ (right) only takes frequency input. The two encoders do not share any weights. The representations of $g_{im}(.)$ are used for downstream tasks. We maximize agreement in the latent space using the standard InfoNCE loss described in Equation 2. Figure 5 illustrates this two-encoder setup.

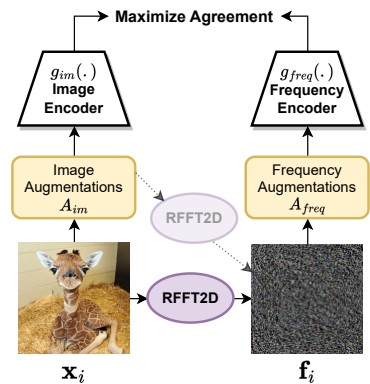

Note that our goal is to disentangle format transforms from augmentations. We take the naive baseline that uses two raw un-augmented views (($\mathbf{x}, \mathbf{x}$) single encoder setup) and substitute the right view with the frequency image and train under the dual encoder regime (Figure 5). In Table 5, we present an interesting finding where contrasting raw image and frequency ($\mathbf{x}, \mathbf{f}$) results in $17.5\%$ top-1 accuracy on ImageNet pre-training which is a $16\%$ improvement over the raw baseline of $1.5\%$. Keeping the left view un-augmented, we augment the right view (i) in the frequency space ($A_{freq}(\mathbf{f})$) which improves the performance to $20.6\%$ and (ii) in the image space before applying Fourier transform ($\mathcal{F}(A_{im}(\mathbf{x}))$) which improves the performance to $48.8\%$.

Next, we augment the left view ($A_{im}(\mathbf{x})$) and contrast against the set of frequency space right views. We do not observe improved performance with format transforms in this scenario. In fact, the performance degrades further when the frequency view is augmented. We hypothesize

Figure 5: **Dual-encoder setup for multi-format contrastive learning:** To disentangle the effect of the format transform, we design a two-encoder setup where the left encoder $g_{im}(.)$ encodes the image view and the right encoder $g_{freq}(.)$ encodes the Fourier transform of the same view. Format-specific augmentations are applied to both views. Both encoders are trained independently (no shared weights) and are aligned in latent space via contrastive loss.

that this behavior may be caused by our choice of architecture for the frequency encoder i.e., ResNet (ConvNets). The *translation equivariance* property of convolutional neural networks that applies to real images, need not directly transfer to frequency images. The improvements we observe from the format transform in lieu of image augmentations in the left view are still non-trivial, opening a new direction for further research.

## 7 Discussion

We examined the need for diverse augmentations in self-supervised pre-training and proposed Frequency-Domain Augmentations (FDA) to introduce further diversity by tapping into the format transform of the image. FDA, when used in conjunction with image augmentations, showed improved performance on ImageNet-1K top-1 accuracy on 4 baselines - SimCLR, BYOL, MoCov2 and SimSiam. We also showed improvements in transfer learning, few-shot learning and image retrieval. We studied the disentangled effect of format transform using a dual-encoder setup with a dedicated frequency encoder. When no augmentations are used, we observed a $16\%$ improvement in performance with the use of format transform in one view as compared to images in both views. Pre-training with the format transform improves over raw images, however, the best performance is still seen in the image space through diverse Fourier (FDA) and image augmentations. Our findings

Table 5: **Disentangling the effect of format transform:** We examine the effect of contrasting image and frequency views using the dual-encoder setup outlined in Figure 5 (cells highlighted in blue). We compare this against the single-encoder setup which uses both image views (first row). When the left image is not augmented, we observe noticeable improvements with format transform (and augmentations) in the right view. We do not observe similar improvements when the left image is augmented.

| Augmentation | | Left View | |
|---|---|---|---|
| | | $\mathbf{x}$ | $A_{im}(\mathbf{x})$ |
| **Right View** | $\mathbf{x}$ | 1.5 | 68.6 |
| | $\mathbf{f}$ | 17.5 | 63.3 |
| | $A_{freq}(\mathbf{f})$ | 20.6 | 62.4 |
| | $\mathcal{F}(A_{im}(\mathbf{x}))$ | 48.8 | 59.0 |

open several questions for further research – (i) What are better methods to utilize and encode the format transform and FDA without requiring to invert back into the image space?, (ii) How can complex Fourier input be better structured to feed through real valued encoders?, (iii) How does FDA behave in specialized domains that are not real images (e.g., medical scans).

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

# A  Appendix

|  | SimCLR | BYOL | MoCov2 | SimSiam |
|---|---|---|---|---|
| Encoder | ResNet-50 | ResNet-50 | ResNet-50 | ResNet-50 |
| Zero init residual | False | False | False | True |
| Projection model features | MLP (4096, 256) | MLP (4096, 256) | MLP (4096, 256) | MLP (2048, 2048, 2048) |
| Prediction model features | N/A | MLP (4096, 256) | N/A | MLP (512, 2048) |
| Momentum encoder (for right encoder) | False | True | True | True |
| Stop-grad (for right encoder) | False | True | True | True |
| Contrastive loss temperature | 0.1 | N/A | 0.1 | N/A |
| Optimizer | LARS | LARS | LARS | LARS |
| Learning rate | 0.2 | 0.2 | 0.2 | 0.2 |
| Weight decay | $1.5 \times 10^{-6}$ | $1.5 \times 10^{-6}$ | $1.5 \times 10^{-6}$ | $1.5 \times 10^{-6}$ |
| Learning rate schedule | cosine decay | cosine decay | cosine decay | cosine decay |
| Epochs | 1000 | 1000 | 1000 | 1000 |
| Linear probe epochs | 90 | 90 | 90 | 90 |
| Linear probe learning rate | 0.3 | 0.3 | 0.3 | 0.3 |
| Linear probe optimizer | SGD | SGD | SGD | SGD |
| Linear probe learning rate schedule | cosine decay | cosine decay | cosine decay | cosine decay |

Table A.1: **Training setup for each model**: We provide the specific architecture and training setup for each encoder for reproducibility.

| Augmentation | Hyper-parameter Values | Probability (Left View) | Probability (Right View) |
|---|---|---|---|
| Random Resized Crop | $224 \times 224$, min area: 0.08, max area: 1.0, min aspect: 3/4, max aspect: 4. / 3., aspect dist: log, resize method: bicubic | 1.0 | 1.0 |
| Color jitter | contrast: 0.4, brightness: 0.4, saturation: 0.2, hue: 0.1 | 0.8 | 0.8 |
| Grayscale | N/A | 0.2 | 0.2 |
| Horizontal flip | N/A | 0.5 | 0.5 |
| Gaussian blur | min sigma: 0.1, max sigma: 2.0, kernel size: 23 | 1.0 | 0.1 |
| Amplitude rescale | $m = 0.8, n = 1.75$ | 0.2 | 0.0 |
| Phase shift | $p = 0.4, q = 0.7$ | 0.2 | 0.0 |
| Random frequency mask | $k \sim [0.01, 0.1)$ | 0.5 | 0.0 |
| Gaussian mixture mask | $c = 20, \sigma \sim [10, 15)$ | 0.2 | 0.0 |

Table A.2: **Augmentation hyperparameters:** We provide the parameters used for each augmentation, both image and FDA along with the probability.

## A.1  Training Setup

We provide all our implementation details for each baseline - SimCLR, BYOL, MoCov2 and SimSiam in Table A.1. We also include linear probing hyperparameters for full reproducibility.

## A.2  Augmentation Hyperparameters

We provide the parameters used for each image and FMA augmentation along with the probability in the left and right view in Table A.2.

