# OpenReview forum: "Disentangling the Effects of Data Augmentation and Format Transform in Self-Supervised Learning of Image Representations"
_NeurIPS.cc/2024/Workshop/UniReps — UniReps_

### Official Review · Reviewer_o23u · 2024-09-29
**Reasonable studies made to support the claim, but is the improvement stat-sig**

**Rating:** 7
**Confidence:** 3

**Review:**

1. What is the reason for the FDA not beating the SimSiam top-1 accuracy in the Places365 dataset?
2. Are the improvements reported in Table 2, statistically significant?
3. From lines 266-267, it appears that applying FDA does not make much of a difference. Can all of this be attributed to the "cropping" that you mention a few lines later?

---

> ### Author Response · Authors · 2024-10-17
>
> We thank Reviewer o23u for accepting our work and for their comments. We respond to them below:
>
> 1. We observe that FDA beats the baselines in most of our transfer experiments. The impact of FDA is also contingent upon the specific model's training strategy and optimization landscape.
> 2. We report an average score across all few-shot and transfer experiments to capture a shared statistic where we see improvements across each model. We could not repeat random seeds due to resource and time limitations however, reporting mean and standard deviation is also not a standard practice in most representation learning works.
> 3. Random cropping is certainly an influential augmentation (similarly observed in Figure 2). The lower accuracies in Table 3 column 3 is due to the fact that there are limited spatial invariances.

---

### Official Review · Reviewer_t2uH · 2024-10-04
**This paper explores the impact of using format transform as a form of augmentation discriminative SSL; This work is well written and covers a relevant topic; The main contribution of this work is the systematic evaluation of format transform augmentation on discriminative SSL thereby leading to a low to mild impact contribution.**

**Rating:** 6
**Confidence:** 4

**Review:**

This paper proposes a systematic evaluation of the use of format transform as augmentations in discriminative SSL. While the idea is not novel, this work aims at providing clean ablations to allow to understand the impact of such augmentation on downstream performance. The authors conclude that the use of format transform is beneficial for downstream performance thereby suggesting modifications in standard augmentation pipelines.

The paper is clear and well-written. While the ablations proposed by the authors are interesting and lead to a clear valuable take-home message, the depth of this work is relatively limited. The use of domain transform augmentation is not new and the authors do not attempt to explain why these transformations are valuable which would have been a bigger contribution. That being said, I believe the topic covered by the authors is interesting to the Unireps community and participates in the ongoing discussion on striving towards 1) more general representations 2) more principled ways to create paired observation for visual representation learning.

---

### Official Review · Reviewer_x5pF · 2024-10-07
**An exploration of data augmentation for SSL in Fourier domain.**

**Rating:** 6
**Confidence:** 4

**Review:**

his articles discusses the importance of diversifying the range of data augmentation techniques for (vision) self-supervised learning and then explores data augmentation in Fourier domain (FDA) in vision, which constitutes the main idea of the paper. While in the audio field, working with a spectral (FFT) version of your signal is a standard technique (e.g., in speech recognition, source separation, etc.) and has been successfully applied to self-supervised learning, it is not widely exploited in the vision SSL field. As such, the authors propose to obtain a FFT of the image, apply spectral augmentations (that they propose) and then revert back to the image domain with the inverse transform, where additional standard augmentations can be applied.  The augmentations are amplitude rescale (which changes colour globally), phase shift (which adds a movement effect globally), random frequency masks (which adds patches of colours) and global gaussian mixtures masks, generalising low and high pass filters. Finally, they also propose feeding the Fourier representations to the encoder directly without passing trough the inverse transform, studying the usage of such representation (and augmented versions of it) as a view, with standard representations as the other view (image domain and augmented image domain). The authors show that Fourier augmentations on the left view improve over standard baselines SimCLR, MoCov2, BYOL and SimSiam (ImageNet-1K), transfer / few-shot learning and disentangle the effect of FDA and image augmentations.

The article is written in a clear way and can be followed easily.

Pros:

- Exploring data augmentation seems natural in Fourier domain.
- The results in Table 5 are pretty interesting: passing from 1% to 17% when using image and spectral views (without data augmentation) is a surprising result and  points that more research has to be done on exploiting spectral representations in vision SSL.

Cons:

- Some sections seem to not add much to the discourse of the paper or are common knowledge in machine learning literature. For example it is well know that (well-defined) additional data augmentations virtually increases the size of your dataset resulting in better model performance so the first claim of the paper feels more like an additional corroboration than a really novel idea. Also the experiments on image retrieval seem a little bit arbitrary since it could be easily cherry picked.

- The improvements obtained by the proposed techniques are present but not very substatial (average of 1%).


Remarks / corrections:

- I do not understand what is the relation between the presented method and the topics of the UniReps conference. Is it the idea that the natural domain and the spectral domain represent the same information in different ways that can be exploited to improve the performance of SSL methods? Isn’t tough this better aimed at the NeuREPS workshop?
- Figure 2: The figure contains everything on the same row as opposed to the description.
- Table 1: I do not understand if the results are obtained by applying FDA only on the left view.
- I wonder why authors do not study both spectral views in Section 6.1.
-  All setups are pre-trained on ImageNet-1K and we report the linear probing top-1 accuracy (Line 240): I find it interesting that you get 30% with x / F^-1, and F^-1 vs F^-1. It still seems that most important factor is still the random cropping in image domain (since SimCLR paper).
- “The two encoders do not share any weights and are trained independently.” (Line 291): It is not clear when say trained independently. At first glance I thought that they were pre-trained stand-alone, then aligned in a second phase.
- In Table 2, use bold on SimSiam +FDA on ImageNet (5-shot) and BYOL + FDA on Food101.
- Line 222: unconventional notation for the argmax.

---

> ### Author Response · Authors · 2024-10-17
>
> We thank Reviewer x5pF for their detailed review and feedback. We respond to some of the remarks below:
>
> - We apply FDA to both views along with standard augmentations for all the results in Table 1
> - We will further investigate using two Fourier views in a dual-encoder setup.
> - Random cropping is certainly an influential augmentation (similarly observed in Figure 2). The ~30% accuracies in Table 3 column 3 is due to the fact that there are limited spatial invariances.
> - Line 291: To clarify, the two encoders are still trained together but do not share any weights. We will change this language in the updated draft.
> We have also incorporated all the suggested corrections in the updated version and we thank the reviewer for bring them to our notice.

---

### Decision · Program_Chairs · 2024-10-10

**Decision:**

Accept

**Comment:**

In light of the positive reviewers' feedback and relevancy of the submission, we are pleased to accept this paper for presentation at UniReps 2024. We kindly ask the authors to incorporate the reviewers' suggestions and feedback in the final camera-ready version of the manuscript.